# Linear Ubiquitin Code: Its Writer, Erasers, Decoders, Inhibitors, and Implications in Disorders

**DOI:** 10.3390/ijms21093381

**Published:** 2020-05-11

**Authors:** Daisuke Oikawa, Yusuke Sato, Hidefumi Ito, Fuminori Tokunaga

**Affiliations:** 1Department of Pathobiochemistry, Graduate School of Medicine, Osaka City University, Osaka 545-8585, Japan; oikawa.daisuke@med.osaka-cu.ac.jp; 2Center for Research on Green Sustainable Chemistry, Tottori University, Tottori 680-8552, Japan; yusato@iam.u-tokyo.ac.jp; 3Department of Neurology, Faculty of Medicine, Wakayama Medical University, Wakayama 641-8510, Japan; itohid@kuhp.kyoto-u.ac.jp

**Keywords:** inflammation, inhibitor, innate immune, interferon, LUBAC, NF-κB, ubiquitin

## Abstract

The linear ubiquitin chain assembly complex (LUBAC) is a ubiquitin ligase composed of the Heme-oxidized IRP2 ubiquitin ligase-1L (HOIL-1L), HOIL-1L-interacting protein (HOIP), and Shank-associated RH domain interactor (SHARPIN) subunits. LUBAC specifically generates the N-terminal Met1-linked linear ubiquitin chain and regulates acquired and innate immune responses, such as the canonical nuclear factor-κB (NF-κB) and interferon antiviral pathways. Deubiquitinating enzymes, OTULIN and CYLD, physiologically bind to HOIP and control its function by hydrolyzing the linear ubiquitin chain. Moreover, proteins containing linear ubiquitin-specific binding domains, such as NF-κB-essential modulator (NEMO), optineurin, A20-binding inhibitors of NF-κB (ABINs), and A20, modulate the functions of LUBAC, and the dysregulation of the LUBAC-mediated linear ubiquitination pathway induces cancer and inflammatory, autoimmune, and neurodegenerative diseases. Therefore, inhibitors of LUBAC would be valuable to facilitate investigations of the molecular and cellular bases for LUBAC-mediated linear ubiquitination and signal transduction, and for potential therapeutic purposes. We identified and characterized α,β-unsaturated carbonyl-containing chemicals, named HOIPINs (HOIP inhibitors), as LUBAC inhibitors. We summarize recent advances in elucidations of the pathophysiological functions of LUBAC-mediated linear ubiquitination and identifications of its regulators, toward the development of LUBAC inhibitors.

## 1. Introduction

Ubiquitin, a 76-residue (8.6 kDa) small globular protein, is evolutionally conserved in most eukaryotes. Ubiquitin functions as a spatiotemporal-specific post-translational modifier. In most cases, ubiquitin is covalently conjugated to the ε-NH_2_ group of Lys in the targeted proteins via an isopeptide bond [1]. In particular cases, ubiquitin is conjugated to the N-terminal α-NH_2_-group of Lys-less proteins and internal Ser, Thr, and Cys residues [2,3]. The human ubiquitin system is composed of two ubiquitin-activating enzymes (E1), ~40 ubiquitin-conjugating enzymes (E2), >600 ubiquitin ligases (E3), and ~100 deubiquitinating enzymes (DUBs). Among them, the E3s play crucial roles in recognizing and conjugating one or more ubiquitins to substrates, and are classified into the homologous to the E6AP carboxyl terminus (HECT)-type, the really interesting new gene (RING)-type, and the RING-in between-RING (RBR)-type [4]. In the human genome, most of the E3s are RING-type [5], and limited numbers of HECT-type (28 members) and RBR-type (14 members) E3s have been identified.

Protein ubiquitination regulates numerous cellular functions, including proteasomal degradation, membrane trafficking, DNA repair, and signal transduction, through the conjugation of one or more ubiquitins to substrates [1]. The conjugation of monoubiquitin and multiple-monoubiquitins to substrates is principally involved in membrane trafficking and endocytosis. The isopeptide bond-linked ubiquitination is mediated via seven internal Lys residues (K6, K11, K27, K29, K33, K48, and K63). Among them, the K48-linked polyubiquitin chain, which is the most common, serves as a typical proteasomal degradation signal, and the K63-linked polyubiquitin chain, the second most predominant linkage, is involved in non-proteasomal functions, such as signal transduction and DNA repair [1,6]. In addition to these Lys-linked polyubiquitin chains, the N-terminal Met1(M1)-linked linear polyubiquitination is specifically generated by the E3 complex, named the linear ubiquitin chain assembly complex (LUBAC). LUBAC functions in the regulation of the innate and acquired immune responses and anti-apoptosis [7]. In addition to the mammalian LUBACs, an ortholog of a HOIP subunit of LUBAC, named LUBEL, was identified in *Drosophila*, and it also catalyzes linear ubiquitination upon heat shock [8]. These findings indicated that linear ubiquitination is evolutionally conserved to maintain cellular homeostasis. In addition to the homotypic polyubiquitin chains, heterogeneous complex-types of polyubiquitin chains, such as mixed, hybrid, and branched ubiquitin chains, have also been identified. Furthermore, specific residues of ubiquitin are chemically modified, by phosphorylation, acetylation, and ADP-ribosylation, and these modifications regulate the pathophysiological functions of ubiquitination. These diverse ubiquitin linkages exhibit multiple functions in a system called the “ubiquitin code” [6], in which E1, E2, and E3 function as “writers”, DUBs are “erasers”, and ubiquitin-binding proteins serve as “decoders”. In this review, we focus on the structures, catalytic mechanisms, inhibitors, and pathophysiological functions of the LUBAC-mediated “linear ubiquitin code”, revealed by studies using human cell lines, diseases, and phenotypes of genetically deficient mice.

## 2. LUBAC: The Only Writer of the Linear Ubiquitin Code

### 2.1. Structure and Catalytic Mechanism of LUBAC

Mammalian LUBAC, a ~600 kDa complex composed of the HOIL-1L (also known as RBCK1) [9], HOIP (RNF31, ZIBRA, and PAUL) [10], and SHARPIN [11,12,13] subunits, is the sole E3 that generates the N-terminal M1-linked linear polyubiquitin chain, using the E2s UBE2L3 (UbcH7) and UbcH5s [14,15]. LUBAC subunits contain multiple domains (Figure 1). Although the detailed architecture of LUBAC has not been solved, the ubiquitin-like (UBL) domains in SHARPIN and HOIL-1L bind to the ubiquitin-associated (UBA)1 and UBA2 domains, respectively, in HOIP [16]. Moreover, the LUBAC-tethering motifs (LTMs) in HOIL-1L and SHARPIN associate with each other to form a globular domain [16]. HOIL-1L and HOIP possess RBR-type E3 motifs. The RBR-type E3 family members reportedly generate polyubiquitin chains through a unique RING-HECT-hybrid reaction [17,18]. During the course of linear ubiquitination, the RING1 domain in HOIP binds a ubiquitin-charged E2. Subsequently, the donor ubiquitin is transiently transferred to the active Cys885 in the RING2 domain of HOIP via a thioester-linkage. Finally, the donor ubiquitin is conjugated to an acceptor ubiquitin, which is captured in the C-terminal linear ubiquitin chain determining domain (LDD) of HOIP, to specifically generate a linear ubiquitin chain [19,20,21,22]. In contrast to HOIP, the RBR domain in HOIL-1L uniquely catalyzes the oxyester-bond monoubiquitination of Ser/Thr residues through the active Cys458 [23].

The N-terminal portion of HOIP contains a PNGase/UBA or UBX (PUB) domain (Figure 1), which is reportedly an AAA-ATPase p97-interacting domain [24] that plays an important role to recruit linear ubiquitin-editing DUBs, such as OTULIN [25] and the CYLD-SPATA2 complex [26,27,28,29]. Thus, LUBAC forms complexes with negative regulators through the PUB domain. Furthermore, LUBAC includes several zinc finger domains (ZFs). Among them, the Npl4-type zinc finger (NZF) domain in HOIL-1L specifically binds linear ubiquitin [30], whereas the NZF domain in SHARPIN binds K63-ubiquitin to regulate the cell death pathway [31]. There are two NZF domains in HOIP, and NZF1 binds NEMO, a LUBAC substrate, during linear ubiquitination [32]. Collectively, these findings indicate that LUBAC consists of multiple functional domains for the regulation of linear ubiquitination and participates in various physiological phenomena.

### 2.2. Cellular Functions of LUBAC

#### 2.2.1. LUBAC in the Inflammatory Cytokine-Induced canonical NF-κB Activation Pathway

LUBAC principally participates in the regulation of the canonical NF-κB signaling pathway in various mammalian cells. NF-κB is a master transcription factor for the biological defense system, and is composed of homo- or hetero-dimers of Rel-homology domain-containing proteins, such as p65 (RelA), RelB, c-Rel, p105/p50, and p100/p52. NF-κB expression leads to the transcription of target genes in the inflammatory and immune responses [33]. LUBAC regulates the NF-κB activation pathways induced by proinflammatory cytokines, such as TNF-α and IL-1β, various pathogen-associated molecular patterns (PAMPs), T cell receptor (TCR) agonists, genotoxic stress, and NOD2-mediated inflammasome activation [14,15]. However, LUBAC is not involved in either the B cell receptor (BCR)-mediated pathway or the noncanonical NF-κB pathway [12,34].

Upon stimulation of cells with TNF-α, LUBAC is recruited to the TNF receptor (TNFR) through binding to K63-linked polyubiquitin chains, which are antecedently generated by c-IAP-1/2, TRAF2, and TRAF5, and functions as a member of the TNFR signaling complex I [35,36,37]. LUBAC then conjugates linear ubiquitin chains to NEMO, RIP1, and FADD (Figure 2) [11,38,39]. The linear ubiquitin chain functions as a scaffold to recruit canonical IκB kinase (IKK) molecules, which are composed of the kinase subunits of IKKα and IKKβ, and a regulatory subunit of NEMO. Importantly, NEMO contains a high-affinity linear ubiquitin binding site that accumulates multiple IKK molecules on the linear ubiquitin chain. The *trans*-phosphorylation of the IKK molecules principally leads to the activation of IKKβ, which subsequently phosphorylates the inhibitory protein of NF-κB, IκBα. Interestingly, the conjugation of two linearly linked molecules of ubiquitin (linear diubiquitin) to NEMO sufficiently induces IKK activation [40]. The phosphorylated IκBα is ubiquitinated by the E3 complex SCF^β-TrCP^ for the K48-ubiquitination-mediated proteasomal degradation of IκBα. After liberation from IκBα, the canonical NF-κB transcription factors, predominantly composed of homo- or hetero-dimers of p65 (RelA) and/or p50, translocate into the nucleus and activate NF-κB target genes (Figure 2) [32]. Upon TNF-α stimulation, mammalian Ste20-like kinase (MST1, also called STK4) is recruited to TNFR in a TRAF2-dependent manner and phosphorylates Ser1066 in the LDD domain of HOIP, which attenuates the E3 activity of LUBAC [41]. Recently, *Parkin-coregulated gene* (PACRG) was identified as a functional replacement of SHARPIN in TNF signaling in human and mouse cells [42]. Therefore, multiple factors regulate the LUBAC-mediated NF-κB activation pathway.

Although IL-1β is another prominent proinflammatory cytokine that activates the canonical NF-κB activation pathway, both K63- and M1-linked ubiquitinations are required for the formation of the NEMO-containing punctate structure upon IL-1β stimulation [43]. Importantly, the K63/M1-hybrid ubiquitin chain can become conjugated to interleukin 1 receptor-associated kinase 1 (IRAK1) and IRAK4 [44]. Furthermore, HOIL-1L conjugates oxyester-bond monoubiquitin to its own Ser/Thr residues, as well as those in SHARPIN, IRAK1/2, and MyD88 in human keratinocyte HaCaT cells and mouse bone marrow-derived macrophages [23]. Thus, the E3 activity of HOIL-1L regulates the Myddosome components upon innate immune responses. These results indicate the differences in the LUBAC functions between the TNF-α- and IL-1β-mediated canonical NF-κB activation pathways.

#### 2.2.2. LUBAC in Acquired Immune Responses

The NF-κB activity plays important roles in lymphocyte development and antigen receptor-mediated acquired immune responses in mammals [33]. Characteristically, a protein complex composed of CARMA1, BCL10, and MALT1 (CBM complex) is critical to activate the B cell receptor (BCR)- and T cell receptor (TCR)-mediated NF-κB activation pathways [45]. In mice B cells, LUBAC has no influence on the IgM-induced BCR pathway, whereas the LUBAC activity is critical for the CD40-mediated NF-κB activation pathway and B1 cell development [34]. In contrast, in T cells, LUBAC is involved in the TCR-mediated NF-κB activation pathway, FOXP3^+^ regulatory T cell (Treg) development, and homeostasis [46]. In the course of the TCR pathway, HOIL-1L is cleaved at Arg165-Gly166 by MALT1, a paracaspase [47]. Moreover, BCL10 is linearly ubiquitinated by LUBAC [48]. However, the importance of the E3 activity of LUBAC in the antigen receptor-mediated NF-κB activation pathway remains to be established [49]. Therefore, further studies are necessary to clarify the function of LUBAC in the antigen receptor-mediated NF-κB activation pathways in lymphocytes.

#### 2.2.3. LUBAC in the Genotoxic Stress Response and Inflammasome Activation

DNA damaging anti-cancer agents, such as camptothecin, etoposide, and doxorubicin, stimulate the NF-κB pathway through the activation of ataxia telangiectasia mutated (ATM) kinase and various post-translational modifications of NEMO, such as phosphorylation, SUMOylation, and ubiquitination [50]. In the genotoxic stress-induced NF-κB activation pathway, X-linked inhibitor of apoptosis (XIAP) conjugates K63-ubiquitin chains to ELKS, which then induces the LUBAC-mediated linear ubiquitination of NEMO in the cytosol [51]. Similarly, the XIAP-mediated K63-linked ubiquitination of RIP2 recruits LUBAC to activate the NOD2-mediated NF-κB activation pathway [52], which plays an important role in the bacterial peptidoglycan-mediated innate immune response.

The inflammasome is a protein complex that activates pro-inflammatory cytokines, such as pro-IL-1β and pro-IL-18. Upon stimulation through Toll-like receptors (TLRs) by damage-associated molecular patterns (DAMPs) and PAMPs, inflammasomes become oligomerized and activate caspase 1. The ubiquitin system functions as both a negative and positive regulator of inflammasomes [53]. The nucleotide binding and leucine-rich repeat-containing protein 3 (NLRP3) is one of the best characterized inflammasomes. LUBAC conjugates a linear ubiquitin chain to the caspase-recruit domain (CARD) of the ASC component, and activates the NLRP3 inflammasome in macrophages [54].

#### 2.2.4. LUBAC-Mediated Regulation of Cell Death

The TNF-α-induced expression of NF-κB-target genes basically functions in anti-apoptosis. However, under conditions where the expression of NF-κB-target genes is suppressed, such as by the protein synthesis inhibitor cycloheximide, TNF-α stimulation extensively induces apoptosis through the generation of TNFR complex IIa, which is composed of RIP1, FADD, and caspase 8 [55] (Figure 2). Subsequently, caspase 8 activates caspase 3 to induce extrinsic apoptosis, and the activated caspases cleave the N-terminal portion of HOIP [39,56]. A genetic deficiency of LUBAC subunits causes reduced expression of NF-κB genes, and thus efficiently induces TNF-α-mediated apoptosis in mice [11,12,13,38,57]. Characteristically, spontaneous *Sharpin*-deficient mice (*cpdm* mice) exhibit severe chronic proliferative dermatitis, and lack secondary lymphoid organs [58,59,60]. The combined genetic deletions of *Tnf* [11] or *Tnfr* [61] with *Sharpin^cpdm^^/cpdm^* mice prevented the skin lesions, indicating that TNF-α-mediated apoptosis plays a critical role in dermatitis in *Sharpin*-deficient mice. Importantly, the lack of secondary lymphoid organs, such as Peyer’s patches, was not alleviated by the attenuation of TNF signaling. However, the genetic deletions of caspase 8 and Rip3k in mice (*Sharpin^cpd/cpdm^/Casp8^+/−^/Rip3k^−/−^*) completely alleviated the phenotype [62]. Recently, Sharma and coworkers showed that the genetic ablation of *MyD88* in *Sharpin*-deficient *cpdm* mice (*Sharpin^cpd/cpdm^/Myd88^−/−^*) completely and partially rescued the skin lesions and systemic inflammation, respectively [63]. Interestingly, they proposed that gut microbiota may play a role in inflammation induction in *cpdm* mice. Therefore, LUBAC seems to be involved in the regulation of not only apoptosis, but also necroptosis.

#### 2.2.5. LUBAC-Mediated Regulation of Interferon Signaling

In innate immune responses, PAMPs are recognized by host pattern-recognition receptors, and then activate the NF-κB and type I interferon (IFN) production pathways. In the course of the IFN antiviral pathway, the phosphorylation of transcription factors, such as interferon regulatory factor 3 (IRF3), by TANK-binding kinase 1 (TBK1) and IKKε causes the transcription of IRF3-target genes, including *IFNβ*, *ISG15*, and *ISG56*. [64]. Although various E3s are known to regulate the antiviral pathway [65], whether LUBAC activates or suppresses the type I IFN production pathway remains controversial. LUBAC and the linear ubiquitination activity reportedly down-regulate the type I IFN production pathway [66,67]. STAT1 is linearly ubiquitinated at the K511 and K652 residues by LUBAC, which inhibits binding to the type I IFN receptor, IFNAR2, and the linear ubiquitination of STAT1 is removed by OTULIN [68]. By contrast, virus infection induces the enhanced LUBAC-mediated NF-κB activation, which in turn, inhibits IFN-STAT1 signaling.

However, some studies suggested that the LUBAC activity is necessary for the TLR-mediated IRF3 activation [69,70,71], and LUBAC is reportedly indispensable for the TNF-induced TBK1 and IKKε activation and prevention of cell death [72]. We showed that the LPS-, poly(I:C)-, and SeV-mediated IFN production pathway was impaired in *HOIP*-deficient mouse embryonic fibroblasts and human Jurkat T-lymphoblasts, and LUBAC inhibitors, HOIPINs, suppressed the antiviral pathway through the reduced activation of TBK1 and IRF3, supporting the positive function of LUBAC in this pathway [73]. Further studies are required to clarify the function of LUBAC in the IFN antiviral pathway.

#### 2.2.6. Involvement of Linear Ubiquitination in Selective Autophagy

Damaged organelles and invading pathogens are selectively degraded through the autophagy pathway [74]. The selective autophagy against intracellular bacteria is referred to as xenophagy. For instance, invaded *Salmonella* replicates within the host-derived membrane vacuole, and the rupture of the vacuole causes ubiquitination and autophagy [75]. The ubiquitinome analysis showed that *Salmonella* infection promotes CDC42 and LUBAC activities with the enhanced NF-κB activity through linear ubiquitination in human colon cancer HCT116 cells and HeLa cells [76]. In addition to several E3s, such as Parkin, Smurf1, RNF166, ARIH1, and LRSAM1, LUBAC indeed restricts *Salmonella* proliferation through linear ubiquitination, which functions to recruit optineurin (OPTN) and NEMO to induce xenophagy and activate NF-κB, respectively [77,78]. Moreover, a linear ubiquitin-specific DUB, OTULIN, antagonizes LUBAC function in the xenophagy of *Salmonella* [79]. These results indicate that linear (de)-ubiquitination is indispensable for the regulation of bacterial pathogenesis and selective autophagy.

## 3. Erasers of the Linear Ubiquitin Code

DUBs function in the biosynthesis, recycling, editing, and cleavage of ubiquitin linkages, and consequently maintain the intracellular free ubiquitin pool [80,81]. The human DUBs are classified into seven subfamilies: ubiquitin-specific protease (USP; 54 members), ubiquitin C-terminal hydrolase (UCH; 4 members), ovarian tumor protease (OTU; 16 members), Josephins (4 members), motif interacting with ubiquitin (MIU)-containing novel DUB (MINDY; 4 members) [82], zinc finger with UFM1-specific peptidase domain protein (ZUFSP; 2 members) [83], and JAB1/MPN/MOV34 metalloenzymes (JAMM/MPN+; 16 members) [80,81]. The USP, UCH, OTU, Josephin, MINDY, and ZFUBP DUBs belong to the Cys protease family, whereas the JAMM/MPN+ family proteins are zinc metalloproteases. Among the DUBs, OTULIN (OTU family), CYLD (USP family), and A20 (OTU family) (Figure 3) are well known DUBs that regulate the LUBAC-mediated NF-κB signaling pathway [84]. In this chapter, we introduce OTULIN and CYLD as “erasers” of the linear ubiquitin code, and classify A20 as a “decoder” (see Section 4.2).

### 3.1. OTULIN

OTULIN (also called FAM105B and Gumby) is an OTU-family DUB with the catalytic Cys129 residue (Figure 3). OTULIN exclusively hydrolyzes the M1-linked peptide bond between ubiquitins, but none of the K-linked isopeptide linkages [85,86,87]. Therefore, OTULIN downregulates the LUBAC-mediated innate immune responses. Interestingly, OTULIN binds to the N-terminal PUB domain of HOIP through the PUB domain-interacting motif (PIM), and the phosphorylation of Tyr56 in the PIM of OTULIN abrogated the HOIP binding [25,88,89]. In humans, genetic mutations in *OTULIN* cause multiple symptoms, such as recurrent fevers, autoantibodies, diarrhea, panniculitis, and arthritis, which are collectively referred to as OTULIN-related autoinflammatory syndrome (ORAS). These mutations deregulate the LUBAC-mediated linear ubiquitination signal [90,91,92], and the *OTULIN*-deficiency causes cell-type-specific LUBAC degradation [85,93]. Interestingly, knock-in mice expressing the active site mutant of Otulin(C129A) showed enhanced cell death through apoptosis and necroptosis, and the increased production of type I IFN, due to the reduced LUBAC activity [94]. Therefore, under physiological conditions, OTULIN prevents the auto-linear ubiquitination of LUBAC, and functions to maintain the LUBAC activity. During cell death, OTULIN regulates the linear ubiquitination of RIP1, which is cleaved at Asp31 by caspase 3. Moreover, the phosphorylation of Tyr56 in OTULIN is increased, and this is counteracted by the dual-specificity phosphatase 14 (DUSP14) during necroptosis [95]. The OTULIN-mediated suppression of hepatocyte apoptosis plays a crucial role in liver pathogeneses, such as hepatitis, fibrosis, and hepatocellular carcinoma [96]. TRIM32, a RING-type E3, conjugates the K63-linked ubiquitin chain to OTULIN to interfere with its interaction with HOIP in human embryonic kidney (HEK) 293 cells [97], and the interaction between TRIM32 and SNX27, which regulates endosome-to-plasma membrane trafficking, has been confirmed [98]. Thus, OTULIN is a crucial pathophysiological regulator in the linear ubiquitin code.

### 3.2. CYLD

CYLD, a USP family DUB (Figure 3), was initially identified as a cylindromatosis tumor suppressor gene in humans [99]. CYLD downregulates the NF-κB activation pathway by hydrolyzing K63-linked ubiquitin chains [100,101]. The genetic deficiency in *CYLD* causes trichoepithelioma. Importantly, CYLD efficiently hydrolyzes the K63-linked ubiquitin chain and the linear ubiquitin chain, but not the K48-chain [102], and thus regulates innate immune signaling [103]. We and another group showed that the catalytic activity of CYLD is indispensable for the downregulation of LUBAC-mediated NF-κB activation [104,105]. Furthermore, we revealed the structural bases of the CYLD USP domain recognition of either K63 or linear diubiquitin [106], and the interaction of a potential CYLD inhibitor, subquinocin [107]. We further identified that mind bomb homolog 2 (MIB2) is an E3 that leads to the proteasomal degradation of CYLD, and therefore MIB2 affects LUBAC-mediated NF-κB activation [108]. Importantly, the USP domain of CYLD binds to the PUB domain of SPATA2, and the PIM in SPATA2 associates with the PUB domain of HOIP (Figure 3) [26,27,28,29]. Recently, a familial variant of *CYLD* with the M719V missense mutation was identified in the induction of frontotemporal dementia (FTD) and amyotrophic lateral sclerosis (ALS), with enhanced K63-deubiquitinating activity and NF-κB suppression [109]. Moreover, this M719V variant of CYLD impairs autophagosome maturation and increases the cytosolic localization of TDP-43. Thus, the up-regulation of CYLD activity is associated with neurodegenerative diseases, whereas the down-regulation of *CYLD* causes cancers.

## 4. Decoders of the Linear Ubiquitin Code

In the ubiquitin code, various types of ubiquitin chains serve as scaffolds to recruit their specific binding proteins, and subsequently, these locally concentrated proteins are responsible for the cellular functions of the ubiquitin code. Therefore, the ubiquitin chain-specific binding proteins are referred to as “decoders”. In the linear ubiquitin code, several protein motifs, such as the UBD in ABIN proteins and the NEMO (UBAN) domain [110], the NZF domain in HOIL-1L [30], and the 7th zinc finger domain in A20 (A20 ZF7) [104,111], have been identified as linear ubiquitin chain-specific binding domains [112]. In this chapter, we summarize the structures and functions of linear ubiquitin-binding proteins.

### 4.1. UBAN Domain-Containing Proteins: NEMO, OPTN, and ABINs

The UBAN domain forms a homodimeric coiled-coil structure, and is present in proteins such as NEMO, OPTN, and A20-binding inhibitors of NF-κB (ABIN), including ABIN-1, ABIN-2, and ABIN-3 [110]. Although the K63-linked and linear diubiquitins adopt similar conformations, the NEMO UBAN domain shows approximately 100-fold higher affinity to the linear ubiquitin chain than the K63-chain. The hydrophobic patches centered at Ile44 and Phe4 of the distal and proximal parts of linear ubiquitin, respectively, are crucial for the interactions with the UBAN domain [113,114]. The NEMO-UBAN domain is critical for the recruitment of the IKK complex onto the linear ubiquitin chain, which induces the canonical NF-κB activation [32]. Missense mutations in the NEMO-UBAN domain in humans cause X-linked anhidrotic ectodermal dysplasia with immunodeficiency (EDA-ID) [115,116]. Therefore, the linear ubiquitin-binding function of NEMO is indispensable for homeostasis.

OPTN, initially identified as a gene responsible for primary open-angle glaucoma (POAG) [117], has a similar domain organization to that of NEMO, although it does not interact with IKKα/β. OPTN is a multifunctional protein that participates in the regulation of the NF-κB and antiviral signaling pathways, vesicular transport, and selective autophagy, such as mitophagy and xenophagy [118]. In 2010, reports demonstrated that genetic mutations in *OPTN* are associated with ALS [119,120], and include an E478G missense mutation in the UBAN domain of human OPTN. We analyzed the effects of the POAG- and ALS-associated mutants of OPTN on the LUBAC- and TNF-α-mediated NF-κB activation in HEK293T cells, and showed that the ALS-associated OPTN mutants lost their ability to suppress NF-κB activation, mainly due to the dysfunction in the UBAN domain of OPTN [121]. The OPTN-UBAN domain strongly bound linear ubiquitin, with a Kd value of 1.0 μM, and the crystal structure of the OPTN-UBAN domain complexed with linear ubiquitin revealed that it shares a similar architecture with that of NEMO [121]. Thus, linear ubiquitin binding by OPTN regulates NF-κB activation and apoptosis, and consequently suppresses ALS.

ABINs were originally identified as inhibitors of NF-κB signaling, and they regulate multiple signal transduction pathways, apoptosis, virus replication, and cancer progression [122]. At present, the crystal structures of the UBAN domains in ABIN-1 and -2 with linear ubiquitin have been solved, and they basically adopt conformations similar to those in NEMO and OPTN [123,124]. Therefore, the ABINs are potential decoders of the linear ubiquitin code, through linear ubiquitin-binding by their UBAN domains.

### 4.2. A20

A20 (also called TNFAIP3) consists of an OTU family DUB domain at the N-terminus, followed by seven zinc finger (ZF) domains (Figure 3) [125]. A20 is an anti-inflammatory protein strongly induced by TNF-α stimulation. Moreover, dysfunctions and polymorphisms of A20 are correlated with various disorders, such as B cell lymphoma, systemic lupus erythematosus (SLE), inflammatory bowel disease, rheumatoid arthritis, and psoriasis. A20 reportedly removes the K63-linked ubiquitin chain from RIP1 by the DUB activity through the OTU domain, and conjugates K48-linked ubiquitin to RIP1 by the E3 activity in the ZF4 domain, leading to the proteasomal degradation of RIP1 [126]. We showed that A20 strongly inhibits the LUBAC-mediated NF-κB activation in a DUB-activity independent manner [104]. Indeed, although A20 hydrolyzes K48- and K63-linked ubiquitin chains, it does not cleave a linear ubiquitin chain. In contrast, the ZF7 domain of A20 is indispensable for the inhibition of LUBAC-mediated NF-κB activation, through specific binding to the linear ubiquitin chain, with a Kd value of 9 μM [104,111]. We solved the crystal structure of human A20 ZF7 with linear ubiquitin, and found that the B cell lymphoma-inducible missense mutations within ZF7 caused the lack of linear ubiquitin-binding. Furthermore, the DUB activity and ZF4 were not necessary for A20-potentiated RIP1-dependent apoptosis, whereas ZF7 is critical in A20 dimerization and intestinal epithelial cell death [127]. Recent studies using knock-in mice revealed that those carrying mutations in ZF7 spontaneously developed arthritis, although mice with mutations of the OTU or ZF4 domain showed no overt inflammatory phenotype [128,129]. Thus, A20 physiologically functions as a decoder, but not as an eraser, in the linear ubiquitin code through the linear ubiquitin-specific binding by ZF7.

## 5. LUBAC-Related Disorders

NF-κB signaling plays pivotal roles in the innate and adaptive immune responses, and anti-apoptosis. Therefore, the impaired NF-κB activation is closely associated with various disorders, such as cancers, inflammatory, autoimmune, and neurodegenerative diseases, and metabolic syndrome [33,130]. In this chapter, we introduce the disorders closely associated with the dysregulation of the linear ubiquitin code.

### 5.1. Genetic Deficiency of LUBAC Subunits and Related Diseases

The *Hoip* [34] or *Hoil-1l* [131] knockout mice are embryonic lethal, while the spontaneous deficiency of *Sharpin* reportedly induces severe dermatitis [60], indicating that the LUBAC activity is crucial for development and homeostasis. In humans, inherited *HOIL-1L* mutations reportedly induce polyglucosan body myopathy with or without immunodeficiency and autoinflammation (Table 1) [132,133,134]. Moreover, the L72P missense mutation in the PUB domain of *HOIP* in a patient with multiorgan autoinflammation, immunodeficiency, amylopectinosis, and systemic lymphangiectasia [135], and another case of *HOIP* deficiency with early-onset immunodeficiency and autoinflammation, but not amylopectinosis and lymphangiectasia [136], were recently identified. These *HOIP* mutations affect the expression of type I IFN regulated genes. Taken together, these results indicate that the LUBAC activity is required to suppress myopathy, systemic inflammation, and immunodeficiency in humans.

### 5.2. Enhanced LUBAC Expression and Cancers

Diffuse large B cell lymphoma (DLBCL) is the most common type of non-Hodgkin’s lymphoma, and is subclassified into the germinal center B cell-like (GCB) and activated B cell-like (ABC) subtypes [137]. Patients with ABC-DLBCL usually have a worse prognosis than those with GCB-DLBCL, and oncogenic mutations in NF-κB signaling proteins, such as in the *CD79B*, *CARMA1*, *MYD88*, and *A20* genes, are associated with ABC-DLBCL. Staudt’s group identified single nucleotide polymorphisms (SNPs) in human *HOIP* that cause the Q622L and Q584H substitutions, which are significantly associated with ABC-DLBCL [138]. Unexpectedly, these replacements, located in the UBA domain of HOIP, enhance the interactions with HOIL-1L and HOIP, resulting in the increased NF-κB activity. The authors developed stapled peptides, an α-helical short peptide with a hydrocarbon cross-link, targeting the HOIP-HOIL-1L interface, and demonstrated the suppressed viability of ABC-DLBCL cells. Indeed, the silencing of *HOIP* also reportedly reduces the viability of ABC-DLBCL cells [49]. Moreover, the E3s of c-IAP-1/2 are involved in ABC-DLBCL via the K63 ubiquitination of BCL10, which results in the recruitment of LUBAC and IKK to the CBM complex, thus inducing BCR-mediated NF-κB activation [139]. Therefore, SMAC mimetics, which lead to the autodegradation of c-IAP-1/2, block the growth of ABC-DLBCL cells. These results indicate that the enhanced LUBAC activity is associated with a poor prognosis in B cell lymphoma (ABC-DLBCL), and thus the inhibition of the LUBAC activity may be a valid therapeutic target.

Recently, Ruiz et al. showed that the expression of LUBAC subunits is enhanced in human and murine lung squamous cell carcinoma (LSCC) cells, but not adenocarcinoma cells, which results in the increased linear ubiquitination, NF-κB activation, and cisplatin-resistance in LSCC [140]. Moreover, the suppression of LUBAC activity, in addition to the suppression of TAK1, ameliorated the chemotherapy resistance of LSCC. Thus, LUBAC inhibitors seem to be therapeutic drug seeds to treat LSCC.

### 5.3. Linear Ubiquitination in Neurodegenerative Diseases

Neurodegenerative diseases, such as Alzheimer’s disease, Parkinson’s disease, Huntington’s disease, ALS, FTD, and related tauopathies, are characterized by the progressive degeneration of the central or peripheral nervous system [141], and at present, no disease-modifying therapies or medicines have been developed. The accumulation of misfolded, aggregated, and ubiquitinated proteins is a common mechanism underlying neurodegenerative diseases. Disease-specific misfolded and aggregated proteins have been identified, such as amyloid-β and tau in Alzheimer’s disease, α-synuclein in Parkinson’s disease, huntingtin in Huntington’s disease, and superoxide dismutase 1 and TAR DNA-binding protein 43 (TDP-43) in ALS [142]. Although the aggregates or inclusions of these proteins are ubiquitin-positive, the types of ubiquitin linkages within these protein aggregates have not been identified.

ALS is a fatal progressive neurodegenerative disease caused by the loss of motor neurons. Although most ALS cases are sporadic, around 10% are familial, and mutations in approximately 20 genes encoding proteins involved in protein/RNA aggregation (*SOD1*, *TDP-43*, *hnRNPA1/2,* and *FUS*), neuroinflammation (*TBK1*), the ubiquitin-proteasome pathway (*UBQLN2*), and autophagy (*C9orf72, OPTN*, *SQSTM1/p62,* and *VCP*) have been identified [143]. Although OPTN reportedly functions as an autophagy receptor, we determined that the ALS-associated *OPTN* mutations, *E478G* and *Q398X*, abrogated the inhibitory effects of OPTN on LUBAC-mediated NF-κB activation, and accelerated TNF-induced cell death in HEK293T and HeLa cells [121]. Importantly, the intracytoplasmic inclusions in neurons from patients with the heterozygous *E478G* and homozygous *Q398X* mutations reacted with an anti-linear ubiquitin antibody, and they were co-localized with TDP-43 and phosphorylated p65, which is an activated form of NF-κB [121]. Moreover, these spinal anterior cells are positive to anti-cleaved caspase 8 and 3 antibodies, suggesting that OPTN regulates neuroinflammation and cell death. We recently showed that the linear ubiquitination of not only the *OPTN*-associated ALS, but also TDP-43-containing inclusions was detected in the cases of sporadic ALS [144]. Interestingly, all of the neuronal cytoplasmic inclusions (NCIs) in spinal cords, including fine “wisps”, were immunolabeled by the anti-K48-linked ubiquitin antibody (Figure 4A,C), whereas the linear ubiquitin was mainly detected in the intermediate and thick bundles of TDP-43-positive inclusions (Figure 4B,E). Similarly, we showed that K63-linked ubiquitin was predominantly detected in intermediate and thick bundles, and linear- and K63-linked ubiquitin immunoreactants were not always co-localized [144]. Moreover, OPTN and phosphorylated p65 were co-localized with these inclusions derived from sporadic ALS patients. Furthermore, we showed that K48-linked ubiquitination is present in all of the tau neurofibrillary tangles, including small ones, whereas a subset of thick neurofibrillary tangles, dystrophic neurites of senile plaques, and neuropil threads were immunopositive for linear ubiquitin in Alzheimer’s disease [145]. These results suggested that various ubiquitinations, including linear ubiquitin, is involved in protein aggregates of neurodegenerative diseases.

Importantly, Winklhofer’s group reported that linear ubiquitin is indispensable for protein quality control [146]. They showed that LUBAC is recruited to the aggregates derived from overexpressed huntingtin-derived polyglutamine (Htt-polyQ) in human neuroblastoma SH-SY5Y cells, and linear ubiquitin also co-localized with the Htt-polyQ aggregates. During the recruitment of LUBAC, AAA-ATPase p97/VCP, which binds to the PUB domain of HOIP, plays an important role to suppress the proteotoxicity, since the linear ubiquitination of Htt-polyQ modulates the aberrant interaction of Htt-polyQ with the transcription factor Sp1. They further indicated that linear ubiquitination is involved in various disease-associated aggregable proteins, such as ataxin-3 (Machado-Joseph disease), and SOD1, TDP-43, and OPTN (ALS). Thus LUBAC, together with its linear ubiquitination activity, seems to be a crucial regulator of various neurodegenerative diseases.

## 6. LUBAC Inhibitors

LUBAC is the sole E3 that can generate a linear ubiquitin chain to regulate acquired and innate immune responses. Therefore, LUBAC inhibitors will facilitate investigations of its enzymatic mechanisms and the cellular bases for immune responses, and serve as potential therapeutics for various LUBAC-related disorders. As described above, stapled peptides targeting the HOIP-HOIL-1L interface suppressed the viability of ABC-DLBCL cells [138]. In this chapter, we summarize the recent advances in the development of small-molecule chemical inhibitors of LUBAC.

### 6.1. Chemical Inhibitors of LUBAC

To date, BAY11-7082 [147], gliotoxin [148], and bendamustine [149] have been reported as chemical inhibitors of LUBAC. However, using non-toxic concentrations, we recently showed that these inhibitors lack the selectivity and inhibitory effects on LUBAC-induced linear ubiquitination and NF-κB activation in HEK293T cells [73].

Recently, Rittinger’s group developed α,β-unsaturated methyl ester-containing compounds, such as compound [5], as LUBAC inhibitors [150,151]. Among them, compound [11a] reportedly inhibited the overexpressed LUBAC-induced NF-κB activity in HEK293T cells (IC_50_ = 37 μM). Importantly, they showed that compound [5] was covalently attached to the catalytic Cys885 of HOIP via Michael addition, by accommodation in a hydrophobic pocket formed by Tyr878, Leu880, and Phe888, and stabilization by hydrogen bonds with the main-chain CO and NH groups of His889 and the Oγ atom of Ser899. These compound [5]-interacting residues of HOIP are located in the RING2 domain [150,151], but are not conserved in other RBR E3s. These sequence variations of RING2 may be beneficial for the HOIP specificity of compound [5].

### 6.2. HOIPINs

To search for novel inhibitors of LUBAC, we constructed a FRET-based assay system to quantify the linear ubiquitination level, using a recombinant LUBAC fragment [152]. After screening 250,000 small molecular chemicals, we identified a thiol-reactive, α,β-unsaturated carbonyl-containing compound, sodium (*E*)-2-(3-(2-methoxyphenyl)-3-oxoprop-1-en-1-yl)benzoate, named HOIPIN-1 from HOIP inhibitor-1, as a LUBAC inhibitor (Figure 5). We developed derivatives of HOIPIN-1, and found that sodium (*E*)-2-(3-(2,6-difluoro-4-(1*H*-pyrazol-4-yl)phenyl)-3-oxoprop-1-en-1-yl)-4-(1-methyl-1*H*-pyrazol-4-yl)benzoate, designated as HOIPIN-8, is the most potent LUBAC inhibitor (Figure 5) [153]. HOIPIN-1 and HOIPIN-8 inhibited the in vitro linear ubiquitination activity of recombinant LUBAC, with IC_50_ values of 2.8 μM and 11 nM, respectively. Furthermore, HOIPIN-1 and HOIPIN-8 suppressed the overexpressed LUBAC-induced NF-κB activity in HEK293T cells, with IC_50_ values of 4.0 μM and 0.42 μM, respectively, indicating that HOIPIN-8 is the most potent among the reported LUBAC inhibitors.

We recently determined that HOIPINs are conjugated to the active site Cys885 in the RING2 domain of HOIP through Michael addition, and interrupt the RING-HECT-hybrid reaction in HOIP [73]. The benzoate, 2,6-difluorophenyl, and 1*H*-pyrazol-4-yl moieties of HOIPIN-8 interact with Arg935 and Asp936, which are located in the LDD domain of human HOIP (Figure 5). Arg935 and Asp936 in the LDD domain reportedly interact with Glu16 and Thr14 in the acceptor ubiquitin, and the Ala mutations of these residues abrogated the linear ubiquitination activity [19,20,21,22]. Thus, HOIPIN-8 not only interacts with the active Cys885, but also masks the critical residues for acceptor ubiquitin-binding by the benzoate, 2,6-difluorophenyl, and 1*H*-pyrazol-4-yl moieties. In addition, the 1-methyl-1*H*-pyrazol-4-yl moiety of HOIPIN-8 interacts with His887, Phe905, and Leu922 of HOIP (Figure 5). These sequence variations of RING2 may be beneficial for the HOIP specificity of compound [5]. On the other hand, the LDD domain of HOIP may be a key determinant for the HOIP specificity of HOIPINs. Therefore, the mechanisms underlying the HOIP specificity are completely different between HOIPINs and compound [5].

Although the α,β-unsaturated carbonyl-containing chemicals seemed to react with various SH-groups, HOIPINs did not inhibit the E1-mediated ubiquitin transfer to E2, or the activities of the HECT-, RING-, and other RBR-type E3s, and specifically suppressed the intracellular linear ubiquitin level induced by TNF-α, IL-1β, and poly(I:C) [73]. As shown in Section 5.2, the enhanced LUBAC activity is associated with the progression of ABC-DLBCL, and we found that HOIPINs effectively suppressed the cell viability of human ABC-DLBCL cells, but not GCB-DLBCL cells, by inhibiting the linear ubiquitination-mediated NF-κB activation [73]. Therefore, the cryptic peptide targeting LUBAC and the HOIPINs may serve as drug seeds to treat ABC-DLBCL patients with a poor prognosis. Moreover, we showed the HOIPIN-1-mediated alleviation of imiquimod-induced psoriasis in model mice. To date, dysfunctions in skin barrier production, IL-23/IL17-mediated lymphocyte signaling, and the NF-κB pathway are reportedly involved in the pathogenesis of psoriasis [154,155]. Since LUBAC affects not only NF-κB but also the production of IL-17 [156], HOIPINs may be effective to treat psoriasis.

## 7. Conclusions and Perspectives

In this review, we have summarized the recent advances in identifying the regulators of LUBAC-mediated linear ubiquitination and its pathophysiological functions. We have highlighted the “linear ubiquitin code”, and its “writer” (LUBAC), “erasers” (DUBs such as OTULIN and CYLD), and “decoders” (linear ubiquitin-binding proteins such as UBAN domain-containing proteins and A20). LUBAC principally activates the canonical NF-κB pathway and suppresses apoptosis. Therefore, the impaired LUBAC activity and the aberrant functions in linear ubiquitin decoders are associated with autoinflammatory and neurodegenerative diseases, and cancers. In particular, it is worthwhile to focus on the fact that linear ubiquitin is present in the protein aggregates of various neurodegenerative diseases, including ALS.

From these studies, we hypothesized that the aggregable proteins in wisp inclusions may be initially conjugated with K48-linked ubiquitin chains; however, they seem to be resistant to the proteasomal degradation due to misfolding (Figure 6). Concomitant with aging, maturation, and liquid–liquid phase separation (LLPS), LUBAC-mediated linear ubiquitin and/or K63-linked ubiquitin chains will be conjugated to NCI. We speculated that it may further generate complex-types of ubiquitin chains, such as K48/linear-branched chains, in the thick bundles. The linear polyubiquitin may serve as a scaffold to recruit the IKK complex and OPTN via the UBAN domain, thus inducing neuroinflammation followed by cell death and selective autophagy, such as mitophagy and aggrephagy (Figure 6). Interestingly, agitation experiments showed that linear polyubiquitin forms fibrillar aggregates more easily than K48-linked polyubiquitin [157]. Therefore, the linear ubiquitination may facilitate the formation of thick inclusions by LLPS. Thus, linear ubiquitination may have pleiotropic effects on neuroinflammation, protein folding, and proteostasis. Although we identified HOIPIN-8 as a potent inhibitor of LUBAC, further studies are necessary to generate potent and specific inhibitors of LUBAC for therapeutic treatments of neurodegenerative diseases.

## Figures and Tables

**Figure 1 ijms-21-03381-f001:**
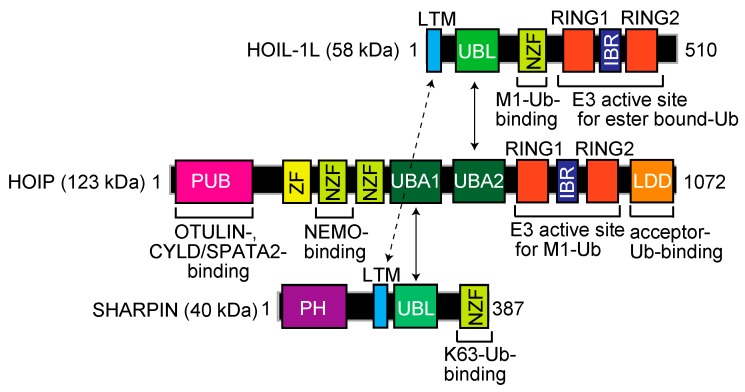
Domain structure and functional regions of the LUBAC subunits, HOIL-1L, HOIP, and SHARPIN. LTM, LUBAC-tethering motif; UBL, ubiquitin-like; NZF, Npl4-type zinc finger; RING, really interesting new gene; IBR, in-between RING; PUB, PNGase/UBA or UBX; ZF, zinc finger; UBA, ubiquitin-associated; LDD, linear ubiquitin chain determining domain; PH, Pleckstrin-homology.

**Figure 2 ijms-21-03381-f002:**
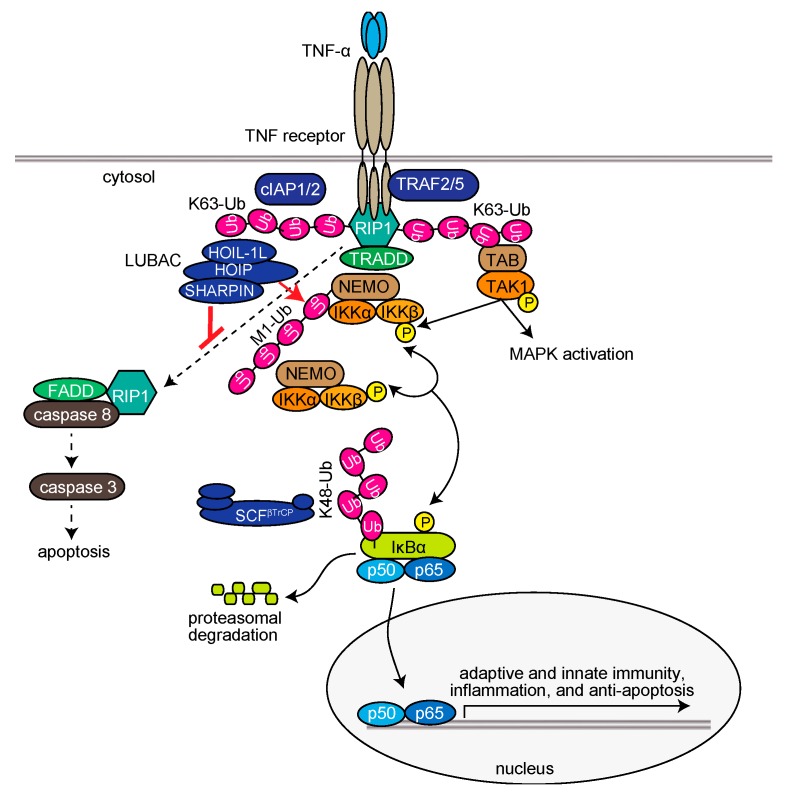
LUBAC-mediated regulation of the TNF-α-induced canonical NF-κB activation pathway and extrinsic apoptosis pathway.

**Figure 3 ijms-21-03381-f003:**
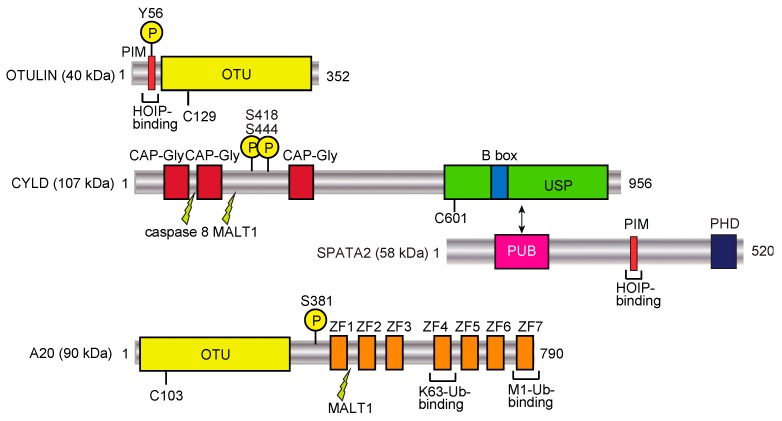
DUBs involved in the regulation of the linear ubiquitin code. Domain structures and functions of LUBAC-regulating factors, such as OTULIN, CYLD-SPATA2, and A20, are shown. PIM, PUB domain-interacting motif; OTU, ovarian tumor protease; CAP-Gly, cytoskeleton-associated protein Gly-rich domain; B box, B-box-type zinc finger domain; PUB, PNGase/UBA or UBX; PHD, plant homeodomain; ZF, zinc finger. Phosphorylation sites are denoted by encircled Ps, and caspase 8 and MALT1 cleavage sites are also indicated.

**Figure 4 ijms-21-03381-f004:**
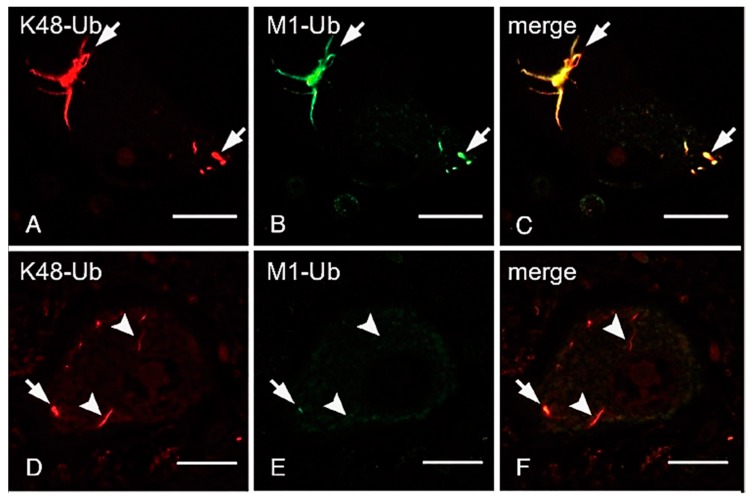
Thick bundles of inclusions are immunopositive for anti-K48 and anti-linear ubiquitin antibodies, whereas fine wisps exclusively reacted with anti-K48 ubiquitin, but not anti-linear ubiquitin. The double immunofluorescence staining using anti-K48-linked ubiquitin antibody (**A**,**D**), anti-linear ubiquitin antibody (**B**,**E**), and their merged images (**C**,**F**) are shown. Arrows; neuronal cytoplasmic inclusions co-localized with K48- and linear-ubiquitins. Arrowheads; K48-ubiquitin-positive inclusions, which lack immunoreactivity for anti-linear ubiquitin antibody (Bars = 20 μm, modified from Ref. [144] with permission).

**Figure 5 ijms-21-03381-f005:**
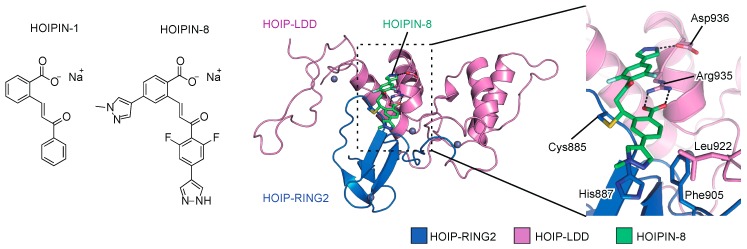
LUBAC inhibitors, HOIPIN-1 and HOIPIN-8. Chemical structures of HOIPIN-1 and -8, and crystal structure of the HOIPIN-8-bound RING2-LDD domain of HOIP are shown.

**Figure 6 ijms-21-03381-f006:**
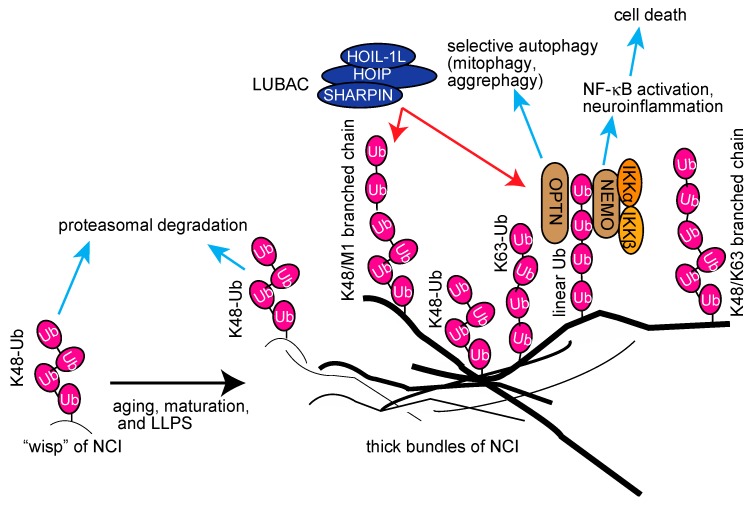
Proposed schema for the involvement of multiple ubiquitin chains, including linear ubiquitin, in neurodegenerative diseases, and their potential functions.

**Table 1 ijms-21-03381-t001:** Genetic deficiencies of human *HOIL-1L* and *HOIP*, and their symptoms.

Gene	Mutations	Symptoms	References
*HOIL-1L*	p.Q185X (c.553C > T), p.L41fsX7 (c.121_122delCT), c.ex1_ex4del	polyglucosan myopathy with immunodeficiency, sepsis, chronic autoinflammation	[132]
	p.Q222X (c.90C > T) p.E190fs (c.68_69insAGGAGCG) c.456+1G > C	polyglucosan body myopathy, cardiomyopathy, muscle atrophy	[133]
	p.E243X (c.727G > T), p.N387S (c.1160A > G), p.E299VfsX18 (c.896_899delAGTG), p.A241GfsX34 (c.722delC), p.A18P (c.52G > C), p.E243GfsX114 (c.727_728ins GGCG), c.ex1_ex4del p.R352X (c.1054C > T) p.R298RfsX40 (c.917+3_917+4insG), p.R165RfsX111 (c.494delG)	polyglucosan myopathy without immunodeficiency	[134]
*HOIP*	p.L72P (c.215T > C)	multiorgan autoinflammation with immunodeficiency, amylopectinosis, systemic lymphangiectasia	[135]
	p.Q399H (c.1197G > C) c.1737+3A > G	eczematous dermatitis, splenomegaly, clubbing of fingers	[136]

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
