# Peer review of "Linear Ubiquitin Code: Its Writer, Erasers, Decoders, Inhibitors, and Implications in Disorders"

_ijms, 2020, doi:10.3390/ijms21093381_

Round 1

Reviewer 1 Report

This is a timely and largely well thought-out review about linear Ub chains, and how they are made, used and disassembled to regulate specific types of processes in the cell. For the first two thirds, the review is a good overview of the studies that have been done. Then it starts to be more confusing and there isn’t a clear delineation between speculations and things that have experimental evidence. Later on, the review seemed to become more about describing the authors' own research and its potential importance, rather than describing research in the field; this made it feel as if there might be something missing from the big picture. The conclusion section is really short; the authors didn’t recenter the ideas to explain how they’re significant or how they could lead to any future therapies for diseases, even though they have listed a lot of diseases throughout. In some ways the review reads like a list of informational pieces. 

This being said, the overall topic will be of importance to the field, and with proper editing and adjustment the review is likely to find a wide audience.

Here is my list of things can than be improved, listed by line number:

  • lines 24-27: As worded, these sentences give the impression that this is a primary research article. may i recommend that the opening sentence of this section be rephrased to make it clear that these inhibitors were identified and characterized in previously published work?
  • lines 237-238: whenever genetic mutations are referred to, or when cell studies are mentioned, it would be helpful to note the animal model and type of cell used: immortalized cells, mice, rats, humans too? flies? c. elegans? this type of detail would be helpful to the reader to further reach their own conclusions.
  • 305-307: as above, wherever possible, it would help to state the type of cells/models used that led to these conclusions.
  • 340: perhaps a table might be generated to summarize all this information about disorders, wherever there are some? it might help the reader to access the information.
  • 387: this entire section was rather difficult to understand. Perhaps the authors might rewrite it to help to clarify the points raised? the section might benefit from being divided into other sub-paragraphs? also, the model proposed for the sequential ubiquitination events in figure 4 seems to come out of observational data, rather than specific experiments that dissect the mechanism. perhaps the authors can be a little clearer about this. 
  • 424-426: a little more discussion would be needed here to help the general reader: are these events of linear ubqn precipitating the disease or secondary or tertiary responses? the nature of the studies summarized here needs to be addressed: cultured cells, neuronal culture, animals? These diseases have immune response pathways that are activated later on. it would help the reader to understand where and how these linear chains and associated proteins might be involved. 
  • 438-443: this part seemed to come to a very abrupt end. perhaps a sentence or two can be added to conclude this part and move onto the next point? additionally, since this inhibitor has such low specificity, is t even worth mentioning in text, or could it just be put into a table, alongside other inhibitors?
  • 495-499: this part felt disjointed from the rest of the section and it did not flow well. Perhaps the authors can provide a better rationale for this information.
  • 501: the conclusion section is extremely limited and does not really synthesize the information presented above. also, proportionally speaking, the conclusions spends more time with ALS than other aspects discussed above. It would be helpful if the authors spent more time synthesizing the overall findings from an immune response aspect of things and brining it all back to ub pathways and types of linkages mentioned in the beginning of the review, and integrating the inhibitors and how they might be used to assess the role of lubac etc in normal immunology response and in, perhaps, neurological diseases.... the authors have done a goods job at detailing information in the proceeding pages. now a bit more time perhaps can be spent with strengthening and shaping the conclusions into a workable set of models or hypotheses that can help pave the way for future work.

Author Response

We appreciate the reviewer’s careful reading and constructive suggestions.

  • lines 24-27: As worded, these sentences give the impression that this is a primary research article. may i recommend that the opening sentence of this section be rephrased to make it clear that these inhibitors were identified and characterized in previously published work?

As recommended by the reviewer, we deleted the following sentence from the Abstract, to avoid overexpression: “Among the HOIPINs, HOIPIN-8 is the most potent inhibitor of LUBAC, by suppressing LUBAC-associated innate immune responses and alleviating the poor prognosis type of B cell lymphoma and psoriasis.”

  • lines 237-238: whenever genetic mutations are referred to, or when cell studies are mentioned, it would be helpful to note the animal model and type of cell used: immortalized cells, mice, rats, humans too? flies? c. elegans? this type of detail would be helpful to the reader to further reach their own conclusions.

In line 242 of the revised manuscript, we rephrased this as “In humans, genetic mutations in OTULIN cause”.

  • 305-307: as above, wherever possible, it would help to state the type of cells/models used that led to these conclusions.

In line 314 of the revised manuscript, we added “in human embryonic kidney (HEK) 293T cells”.

  • 340: perhaps a table might be generated to summarize all this information about disorders, wherever there are some? it might help the reader to access the information.

We thank the reviewer for this comment. In the revised manuscript, we summarized the genetic mutations in the LUBAC subunits of HOIL-1L and HOIP and their resulting symptoms, identified so far, in Table 1.

  • 387: this entire section was rather difficult to understand. Perhaps the authors might rewrite it to help to clarify the points raised? the section might benefit from being divided into other sub-paragraphs? also, the model proposed for the sequential ubiquitination events in figure 4 seems to come out of observational data, rather than specific experiments that dissect the mechanism. perhaps the authors can be a little clearer about this. 

To better understand chapter 5.3, we rewrote the chapter and included the new Figure 4. We believe that the revision clarifies the relationship between pathogenic mutations in neurodegenerative diseases and their cellular functions. By presenting the experimental findings in the new Figure 4, the scheme shown in Figure 5 would be luculent.

  • 424-426: a little more discussion would be needed here to help the general reader: are these events of linear ubqn precipitating the disease or secondary or tertiary responses? the nature of the studies summarized here needs to be addressed: cultured cells, neuronal culture, animals? These diseases have immune response pathways that are activated later on. it would help the reader to understand where and how these linear chains and associated proteins might be involved. 

In lines 440-450 of the revised manuscript, we included detailed and concrete descriptions of Winklhofer’s study, as suggested by the reviewer.

  • 438-443: this part seemed to come to a very abrupt end. perhaps a sentence or two can be added to conclude this part and move onto the next point? additionally, since this inhibitor has such low specificity, is t even worth mentioning in text, or could it just be put into a table, alongside other inhibitors?

We thank the reviewer for this comment. As shown in lines 463-466 of the revised manuscript, we simplified the descriptions of BAY11-7082, gliotoxin, and bendamustine to avoid confusion for the readers, since we do not think these chemicals are LUBAC-specific inhibitors.

  • 495-499: this part felt disjointed from the rest of the section and it did not flow well. Perhaps the authors can provide a better rationale for this information.

By dividing the effects of HOIPINs on ABC-DLBCL and psoriasis, we tried to provide a better rationale in lines 507-516 of the revised manuscript.

  • 501: the conclusion section is extremely limited and does not really synthesize the information presented above. also, proportionally speaking, the conclusions spends more time with ALS than other aspects discussed above. It would be helpful if the authors spent more time synthesizing the overall findings from an immune response aspect of things and brining it all back to ub pathways and types of linkages mentioned in the beginning of the review, and integrating the inhibitors and how they might be used to assess the role of lubac etc in normal immunology response and in, perhaps, neurological diseases.... the authors have done a goods job at detailing information in the proceeding pages. now a bit more time perhaps can be spent with strengthening and shaping the conclusions into a workable set of models or hypotheses that can help pave the way for future work.

We thank the reviewer for this critical comment. In lines 519-533, we rewrote the Conclusion to describe the review comprehensively.

Reviewer 2 Report

The manuscript is extremely interesting, well organized and potentially suitable for publication. It provides a clear and elegant overview of an intriguing topic that still needs further insights. Nevertheless, few more pertinent and up-to-date details could refine the accuracy of the review 

1) In the introduction it could be worth at least to mention the remarkable discovery made by Dikic group in 2016 about Serine ubiquitination (doi.org/10.1016/j.cell.2016.11.019).

2) In the introduction, reference to Asaoka et al. publication on EMBO Rep (doi.org/10.15252/embr.201642378) will strengthen the value of linear ubiquitination mechanism in cell homeostasis.

3) In line 110, in the context of LUBAC involvement in TNF-a stimulation pathway, the authors should mention also the pioneering studies of Ea et al. in 2006 and Haas et al., in 2009 (respectively doi.org/10.1016/j.molcel.2006.03.026, and doi.org/10.1016/j.molcel.2009.10.013).

4) In line 183, referring to SHARPIN-deficient mice skin lesions, it has been recently demonstrated a rescuing effect upon MyD88 deficiency that is worth to note (doi.org/10.1038/s41418-018-0159-7). Also, report of the newly described E3 ligase MIB2 role in CYLD degradation, and its impact on inflammation through LUBAC (doi: 10.1074/jbc.RA119.010119) would perfectly fit in 3.2 CYLD paragraph.

5) Since paragraph 5.1 refers specifically to the LUBAC deficiency related-diseases, it could be worth to mention Ruiz et al. study on LUBAC-mediated chemotherapy resistance in squamous cell lung cancer (doi: 10.1084/jem.20180742).

6)  The implementation of the paragraph 5.1 with a TABLE that relates each genetic mutation to the corresponding diseases would increase the readability of the manuscript.  

6) A minor point regards a misprint at line 341 (10ignalling). 

Author Response

We sincerely thank the reviewer for their helpful comments, especially for the recommendation of adequate citations.

  • In the introduction it could be worth at least to mention the remarkable discovery made by Dikic group in 2016 about Serine ubiquitination (doi.org/10.1016/j.cell.2016.11.019).

In lines 34-35: To introduce the non-canonical ubiquitination we added the following sentence “In particular cases, ubiquitin is conjugated to the N-terminal α-NH2-group of Lys-less proteins and internal Ser, Thr, and Cys residues [2, 3].” and cited the indicated article and a review on non-canonical ubiquitylation.

  • In the introduction, reference to Asaoka et al. publication on EMBO Rep (doi.org/10.15252/embr.201642378) will strengthen the value of linear ubiquitination mechanism in cell homeostasis.

As indicated, we added the following sentences in lines 54-56 and cited the indicated reference: “In addition to the mammalian LUBACs, an ortholog of a HOIP subunit of LUBAC, named LUBEL, was identified in Drosophila, and it also catalyzes linear ubiquitination upon heat shock [8]. These findings indicated that linear ubiquitination is evolutionally conserved to maintain cellular homeostasis.”

  • In line 110, in the context of LUBAC involvement in TNF-a stimulation pathway, the authors should mention also the pioneering studies of Ea et al. in 2006 and Haas et al., in 2009 (respectively doi.org/10.1016/j.molcel.2006.03.026, and doi.org/10.1016/j.molcel.2009.10.013).

In line 113 of the revised manuscript, we cited the indicated references as Ref. 36 and 37, and added the following phrase: “, and functions as a member of the TNFR signaling complex I”.

  • In line 183, referring to SHARPIN-deficient mice skin lesions, it has been recently demonstrated a rescuing effect upon MyD88 deficiency that is worth to note (doi.org/10.1038/s41418-018-0159-7). Also, report of the newly described E3 ligase MIB2 role in CYLD degradation, and its impact on inflammation through LUBAC (doi: 10.1074/jbc.RA119.010119) would perfectly fit in 2 CYLDparagraph. 

As indicated by the reviewer, in lines 186-189, we added the sentence that “Recently, Sharma and coworkers showed that the genetic ablation of MyD88 in Sharpin-deficient cpdm mice (Sharpincpd/cpdm/Myd88-/-) completely and partially rescued the skin lesions and systemic inflammation, respectively [63]. Interestingly, they proposed that gut microbiota may play a role in inflammation induction in cpdm mice.”

We thank the reviewer for the comment on the MIB2 paper. We added the following sentence in line 268: “We further identified that mind bomb homolog 2 (MIB2) is an E3 that leads to the proteasomal degradation of CYLD, and therefore MIB2 affects LUBAC-mediated NF-κB activation [108].”

  • Since paragraph 5.1refers specifically to the LUBAC deficiency related-diseases, it could be worth to mention Ruiz et al. study on LUBAC-mediated chemotherapy resistance in squamous cell lung cancer (doi: 10.1084/jem.20180742). 

We are grateful for the reviewer’s thoughtful comment. Since Ruiz et al. showed that the enhanced expression of LUBAC subunits is associated with lung squamous cell carcinoma (LSCC), we changed the title of chapter 5.2 from “LUBAC and B cell lymphoma” to “Enhanced LUBAC expression and cancers”. Then, with citing the indicated reference, we added the following sentences in line 385, “Recently, Ruiz et al. showed that the expression of LUBAC subunits is enhanced in lung squamous cell carcinoma (LSCC) cells, but not adenocarcinoma cells, which results in the increased linear ubiquitination, NF-κB activation, and cisplatin-resistance in LSCC [140]. Moreover, the suppression of LUBAC activity, in addition to the suppression of TAK1, ameliorated the chemotherapy resistance of LSCC. Thus, LUBAC inhibitors seem to be therapeutic drug seeds to treat LSCC.”

  • The implementation of the paragraph 5.1with a TABLE that relates each genetic mutation to the corresponding diseases would increase the readability of the manuscript.  

We thank the reviewer for this comment. In the revised manuscript, we summarized the genetic mutations in the LUBAC subunits of HOIL-1L and HOIP and their resulting symptoms, identified so far, in Table 1.

7) A minor point regards a misprint at line 341 (10ignalling). 

We apologize for the typographical error. We corrected it to “NF-κB signaling” in line 349 of the revised manuscript.

Round 2

Reviewer 1 Report

I appreciate the comments from the reviewer. However, the points raised in the prior version have been only minimally addressed. List below. Also, i am attaching a PDF with additional comments. So, the authors should address BOTH the points summarized below, as well as the points in the attached PDF

We appreciate the reviewer’s careful reading and constructive suggestions.

  • lines 24-27: As worded, these sentences give the impression that this is a primary research article. may i recommend that the opening sentence of this section be rephrased to make it clear that these inhibitors were identified and characterized in previously published work?

As recommended by the reviewer, we deleted the following sentence from the Abstract, to avoid overexpression: “Among the HOIPINs, HOIPIN-8 is the most potent inhibitor of LUBAC, by suppressing LUBAC-associated innate immune responses and alleviating the poor prognosis type of B cell lymphoma and psoriasis.”

REVIEWER'S REPLY: THANKS

  • lines 237-238: whenever genetic mutations are referred to, or when cell studies are mentioned, it would be helpful to note the animal model and type of cell used: immortalized cells, mice, rats, humans too? flies? c. elegans? this type of detail would be helpful to the reader to further reach their own conclusions.

In line 242 of the revised manuscript, we rephrased this as “In humans, genetic mutations inOTULIN cause”.

REVIEWER'S REPLY: This note was meant not only for this line, but EVERYWHERE in the article where data are described. I would greatly appreciate it if the authors would indicate throughout the paper the models etc used for the experiments described. 

  • 305-307: as above, wherever possible, it would help to state the type of cells/models used that led to these conclusions.

In line 314 of the revised manuscript, we added “in human embryonic kidney (HEK) 293T cells”.

REVIEWER'S REPLY: This note was meant not only for this line, but EVERYWHERE in the article where data are described. I would greatly appreciate it if the authors would indicate throughout the paper the models etc used for the experiments described. 

  • 340: perhaps a table might be generated to summarize all this information about disorders, wherever there are some? it might help the reader to access the information.

We thank the reviewer for this comment. In the revised manuscript, we summarized the genetic mutations in the LUBAC subunits of HOIL-1L and HOIP and their resulting symptoms, identified so far, in Table 1.

REVIEWER'S REPLY: Thanks

  • 387: this entire section was rather difficult to understand. Perhaps the authors might rewrite it to help to clarify the points raised? the section might benefit from being divided into other sub-paragraphs? also, the model proposed for the sequential ubiquitination events in figure 4 seems to come out of observational data, rather than specific experiments that dissect the mechanism. perhaps the authors can be a little clearer about this. 

To better understand chapter 5.3, we rewrote the chapter and included the new Figure 4. We believe that the revision clarifies the relationship between pathogenic mutations in neurodegenerative diseases and their cellular functions. By presenting the experimental findings in the new Figure 4, the scheme shown in Figure 5 would be luculent.

REVIEWER'S REPLY: I don’t understand how figure 4 helps clarify things. In fact, this figure doesn’t show what the authors purport it does; they talk about M1 and K63 chains sometimes co-localizing, and they reference figure 4 for that, but figure 4 doesn’t show K63 staining. This entire section is confusing - the authors write about several different proteins that are in the inclusions, and it’s quite difficult to keep track of it all and to make sense of the overall message. This part of the paper clearly needs a lot of editing and rewriting to be accessible.

This issue spills over into figure 5: the authors talk about how M1 ubiquitination might cause “wisps” to aggregate more and become thicker inclusions, but they said they never see M1 in wisps, and they don’t mention anything about seeing intermediate species. It is very difficult to understand how the authors are able to jump from figure 4 to all of the assumptions made in figure 5.

Additionally, it seems unwarranted to have branched chains depicted in figure 5 when they seem to have only found that the inclusions contain different types of chains, but not necessarily that they are conjugated to each other. If their previous papers clarify any of this, it would be a lot more helpful if figure 4 were a summary of their findings, since it feels like they’re basing figure 5 almost entirely on their own research.

  • 424-426: a little more discussion would be needed here to help the general reader: are these events of linear ubqn precipitating the disease or secondary or tertiary responses? the nature of the studies summarized here needs to be addressed: cultured cells, neuronal culture, animals? These diseases have immune response pathways that are activated later on. it would help the reader to understand where and how these linear chains and associated proteins might be involved. 

In lines 440-450 of the revised manuscript, we included detailed and concrete descriptions of Winklhofer’s study, as suggested by the reviewer.

REVIEWER'S REPLY: Thanks

  • 438-443: this part seemed to come to a very abrupt end. perhaps a sentence or two can be added to conclude this part and move onto the next point? additionally, since this inhibitor has such low specificity, is t even worth mentioning in text, or could it just be put into a table, alongside other inhibitors?

We thank the reviewer for this comment. As shown in lines 463-466 of the revised manuscript, we simplified the descriptions of BAY11-7082, gliotoxin, and bendamustine to avoid confusion for the readers, since we do not think these chemicals are LUBAC-specific inhibitors.

REVIEWER'S REPLY: Thanks

  • 495-499: this part felt disjointed from the rest of the section and it did not flow well. Perhaps the authors can provide a better rationale for this information.

By dividing the effects of HOIPINs on ABC-DLBCL and psoriasis, we tried to provide a better rationale in lines 507-516 of the revised manuscript.

REVIEWER'S REPLY: Thanks

  • 501: the conclusion section is extremely limited and does not really synthesize the information presented above. also, proportionally speaking, the conclusions spends more time with ALS than other aspects discussed above. It would be helpful if the authors spent more time synthesizing the overall findings from an immune response aspect of things and brining it all back to ub pathways and types of linkages mentioned in the beginning of the review, and integrating the inhibitors and how they might be used to assess the role of lubac etc in normal immunology response and in, perhaps, neurological diseases.... the authors have done a goods job at detailing information in the proceeding pages. now a bit more time perhaps can be spent with strengthening and shaping the conclusions into a workable set of models or hypotheses that can help pave the way for future work.

We thank the reviewer for this critical comment. In lines 519-533, we rewrote the Conclusion to describe the review comprehensively.

REVIEWER'S REPLY: The changes made to the conclusion are minimal and do not address my prior comment. I would appreciate it if the reviewers would attempt to fully rewrite the conclusion as a more substantial part of the paper, rather than just a simple coda. 

Author Response

  • lines 237-238: whenever genetic mutations are referred to, or when cell studies are mentioned, it would be helpful to note the animal model and type of cell used: immortalized cells, mice, rats, humans too? flies? c. elegans? this type of detail would be helpful to the reader to further reach their own conclusions.

In line 242 of the revised manuscript, we rephrased this as “In humans, genetic mutations in OTULIN cause”.

REVIEWER'S REPLY: This note was meant not only for this line, but EVERYWHERE in the article where data are described. I would greatly appreciate it if the authors would indicate throughout the paper the models etc used for the experiments described. 

  • 305-307: as above, wherever possible, it would help to state the type of cells/models used that led to these conclusions.

In line 314 of the revised manuscript, we added “in human embryonic kidney (HEK) 293T cells”.

REVIEWER'S REPLY: This note was meant not only for this line, but EVERYWHERE in the article where data are described. I would greatly appreciate it if the authors would indicate throughout the paper the models etc used for the experiments described. 

According the reviewer’s comments, at the end of Introduction we described that “In this review, we focus on…, revealed by studies using human cell lines, diseases, and phenotypes of genetically deficient mice.” Moreover, as shown in the red letters, we added Mammalian (line 70) in various mammalian cells (line 104), in human and mouse cells (line 129), in the human keratinocyte HaCaT cells and mouse bone marrow-derived macrophages (line 136), in mammals (line 145), mice (line 147), in mice (line 180), mouse embryonic fibroblasts and human Jurkat T-lymphoblasts (line 207), in human colon cancer HCT116 cells and HeLa cells (line 217), in human embryonic kidney (HEK) 293 cells (line 258), in human (line 263), in human (line 306), human (line 314), human (line 341), intestinal epithelial (line 344), human (line 375), human and murine (line 387), in HEK293T and HeLa cells (line 411), in spinal cords (line 418), human (line 483), and human (line 500).

In this review article, we have clearly designated the human gene symbols by upper-case Latin letters or by a combination of upper-case letters and Arabic numerals, according to the HUGO recommendations, and as indicated by MGI, we described the mouse gene symbols beginning with an upper-case followed by all lower-case letters. Moreover, we showed the gene names in italics, and the protein names in plain text, as indicated by the nomenclature. Therefore, we believe that the readers will be able to distinguish whether the name represents a human or mouse gene or protein. We sincerely thank the reviewer for evaluating our work with the highest score for the criteria on “appropriate and adequate references to related and previous work”. This is a review article, and not an original paper, and we summarized the previous work within a couple of sentences. In most cases, the studies were done under multiple experimental conditions by using several human and mouse cell lines, but not a specific species or cell type. Therefore, we believe that these revisions will be sufficient for this review, and further detailed information can certainly be obtained from the cited references.

  • 387: this entire section was rather difficult to understand. Perhaps the authors might rewrite it to help to clarify the points raised? the section might benefit from being divided into other sub-paragraphs? also, the model proposed for the sequential ubiquitination events in figure 4 seems to come out of observational data, rather than specific experiments that dissect the mechanism. perhaps the authors can be a little clearer about this. 

To better understand chapter 5.3, we rewrote the chapter and included the new Figure 4. We believe that the revision clarifies the relationship between pathogenic mutations in neurodegenerative diseases and their cellular functions. By presenting the experimental findings in the new Figure 4, the scheme shown in Figure 5 would be luculent.

REVIEWER'S REPLY: I don’t understand how figure 4 helps clarify things. In fact, this figure doesn’t show what the authors purport it does; they talk about M1 and K63 chains sometimes co-localizing, and they reference figure 4 for that, but figure 4 doesn’t show K63 staining. This entire section is confusing - the authors write about several different proteins that are in the inclusions, and it’s quite difficult to keep track of it all and to make sense of the overall message. This part of the paper clearly needs a lot of editing and rewriting to be accessible.

This issue spills over into figure 5: the authors talk about how M1 ubiquitination might cause “wisps” to aggregate more and become thicker inclusions, but they said they never see M1 in wisps, and they don’t mention anything about seeing intermediate species. It is very difficult to understand how the authors are able to jump from figure 4 to all of the assumptions made in figure 5. 

Additionally, it seems unwarranted to have branched chains depicted in figure 5 when they seem to have only found that the inclusions contain different types of chains, but not necessarily that they are conjugated to each other. If their previous papers clarify any of this, it would be a lot more helpful if figure 4 were a summary of their findings, since it feels like they’re basing figure 5 almost entirely on their own research.

  • 501: the conclusion section is extremely limited and does not really synthesize the information presented above. also, proportionally speaking, the conclusions spends more time with ALS than other aspects discussed above. It would be helpful if the authors spent more time synthesizing the overall findings from an immune response aspect of things and brining it all back to ub pathways and types of linkages mentioned in the beginning of the review, and integrating the inhibitors and how they might be used to assess the role of lubac etc in normal immunology response and in, perhaps, neurological diseases.... the authors have done a goods job at detailing information in the proceeding pages. now a bit more time perhaps can be spent with strengthening and shaping the conclusions into a workable set of models or hypotheses that can help pave the way for future work.

We thank the reviewer for this critical comment. In lines 519-533, we rewrote the Conclusion to describe the review comprehensively.

REVIEWER'S REPLY: The changes made to the conclusion are minimal and do not address my prior comment. I would appreciate it if the reviewers would attempt to fully rewrite the conclusion as a more substantial part of the paper, rather than just a simple coda. 

To clarify these points raised by the reviewer, we described the published results in Chapter 5.3. As pointed out by the reviewer, we showed the immunofluorescent staining of K48- and linear-ubiquitin in Fig. 4, but not the K63-linked ubiquitin chain. We think it is important to point out that linear ubiquitin is indeed detectable in the protein aggregates. Therefore, we divided this information into two sentences as shown in lines 419-423, that “whereas the linear ubiquitin was mainly detected in the intermediate and thick bundles of TDP-43-positive inclusions (Fig. 4B, E). Similarly, we showed that K63-linked ubiquitin was predominantly detected in intermediate and thick bundles, and linear- and K63-linked ubiquitin immunoreactants were not always co-localized [144].” We think it is important to point out that the various ubiquitinations of the aggregates were defective in ALS and AD, but they differed in the size of the aggregates. In addition, we referred to the intermediate species.

Since the previous Fig. 5 showed our working hypothesis as described, we moved the paragraph at the end of Chapter 7 with the indication of Fig. 6, and changed the title of the chapter from “Conclusions” to “Conclusions and perspectives”. We tried to rewrite the section to indicate that we are describing a hypothesis to be clarified in the future.

Together, these changes distinguish the experimental results (Chapter 5.3.) and the workable set of models or hypotheses for future work (Chapter 7) to investigate involvement of the linear ubiquitin code in neurodegenerative diseases, as requested by the reviewer.

Round 3

Reviewer 1 Report

The authors have address all the comments.